# Cisplatin is retained in the cochlea indefinitely following chemotherapy

Andrew M. Breglio [1,2,3], Aaron E. Rusheen [1,7], Eric D. Shide[4], Katharine A. Fernandez [1], Katie K. Spielbauer[1], Katherine M. McLachlin[5], Matthew D. Hall [6], Lauren Amable [4] & Lisa L. Cunningham [1]

Cisplatin chemotherapy causes permanent hearing loss in 40–80% of treated patients. It is unclear whether the cochlea has unique sensitivity to cisplatin or is exposed to higher levels of the drug. Here we use inductively coupled plasma mass spectrometry (ICP-MS) to examine cisplatin pharmacokinetics in the cochleae of mice and humans. In most organs cisplatin is detected within one hour after injection, and is eliminated over the following days to weeks. In contrast, the cochlea retains cisplatin for months to years after treatment in both mice and humans. Using laser ablation coupled to ICP-MS, we map cisplatin distribution within the human cochlea. Cisplatin accumulation is consistently high in the stria vascularis, the region of the cochlea that maintains the ionic composition of endolymph. Our results demonstrate long-term retention of cisplatin in the human cochlea, and they point to the stria vascularis as an important therapeutic target for preventing cisplatin ototoxicity.

[1] National Institute on Deafness and Other Communication Disorders, National Institutes of Health, Bethesda, MD 20892, USA. [2] Department of Physiology, Anatomy and Genetics, University of Oxford, Oxford OX1 3QX, UK. [3] Icahn School of Medicine at Mount Sinai, New York, NY 10029, USA. [4] National Institute on Minority Health and Health Disparities, National Institutes of Health, Bethesda, MD 20892, USA. [5] Electro Scientific Industries, Inc., Bozeman, MT 59715, USA. [6] National Center for Advancing Translational Sciences, National Institutes of Health, Rockville, MD 20850, USA. [7] Present address: Medical Scientist Training Program, Mayo Clinic School of Medicine, Rochester, MN 55905, USA. Lauren Amable and Lisa L. Cunningham contributed equally to this work. Correspondence and requests for materials should be addressed to L.L.C. (email: lisa.cunningham@nih.gov)

Cisplatin chemotherapy has been a mainstay of cancer treatment since its FDA approval in 1978. As the population of cancer survivors continues to grow, so does the importance of addressing the long-term sequelae of cancer treatment[1]. Following cisplatin chemotherapy, 40–80% of adults and at least 50% of children are left with permanent hearing loss[2–7]. This hearing loss can result in a multifaceted decrease in quality of life, and in pediatric patients it can impact social and academic development[4,5,8].

Addressing cisplatin ototoxicity requires an improved understanding of why the cochlea is particularly susceptible to this drug. Cisplatin cytotoxicity is thought to be mediated primarily through DNA crosslinking, as well as reactive oxygen species production following binding to cytoplasmic proteins and biomolecules[9–11]. It has previously been assumed that cochlear cell types are particularly sensitive to cisplatin cytotoxicity, and

much experimental focus has been upon identifying the cellular and molecular pathways responsible for this sensitivity[12]. We instead hypothesized that differences in cisplatin pharmacokinetics among organs may account for the unique susceptibility of the cochlea to cisplatin damage.

Here we use inductively coupled plasma mass spectrometry (ICP-MS)[13,14] to quantify biodistribution of platinum metal in murine tissues and human cochleae following cisplatin administration. This approach allows for detection of all active species derived from cisplatin with high sensitivity and no background, since platinum is not present in normal biological tissues. Using this approach, we demonstrate persistence of cisplatin species in the cochleae of experimental mice and human patients for months-to-years following cisplatin chemotherapy. In the mouse, we use both ICP-MS and a fluorescent cisplatin analog to demonstrate high cisplatin accumulation in the stria vascularis

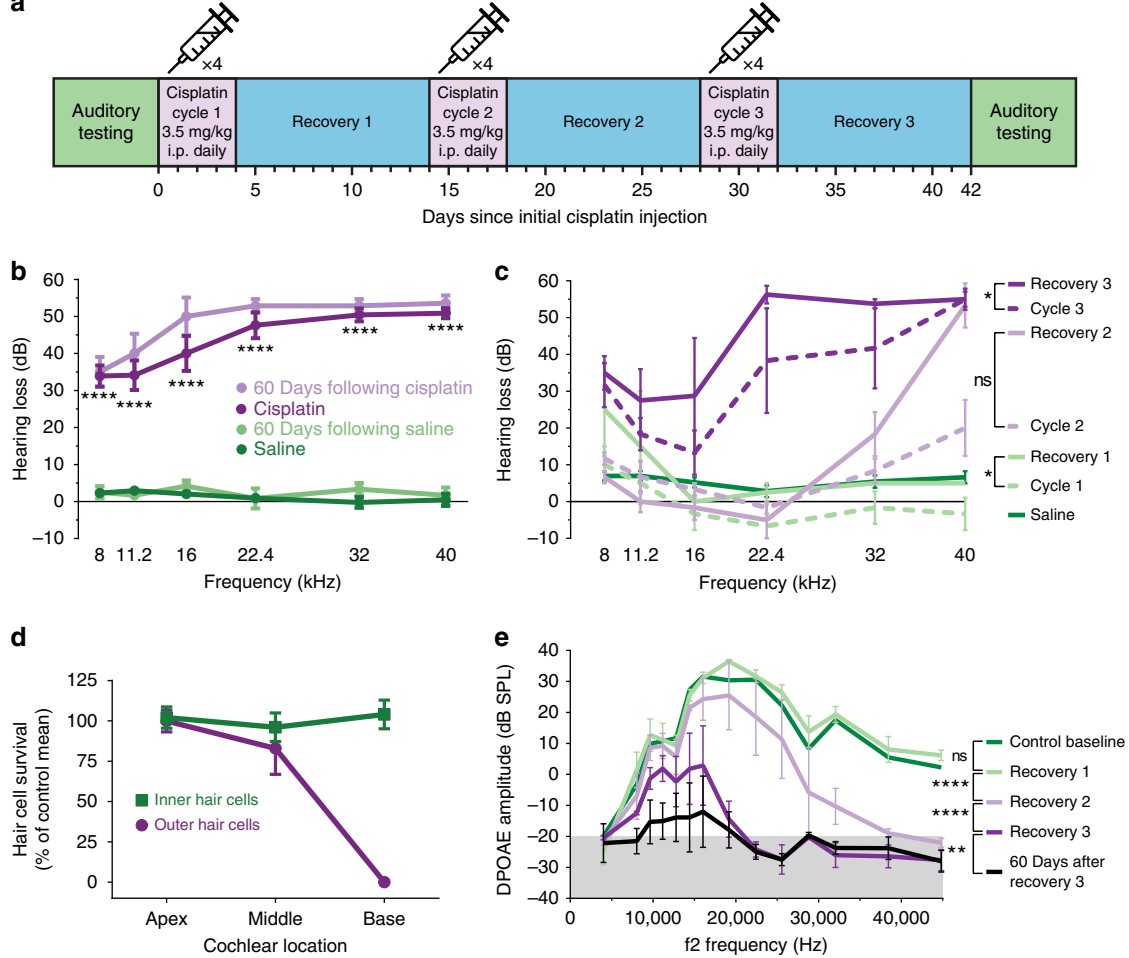

**Fig. 1** A clinically relevant mouse model of cisplatin ototoxicity shows progressive, high-frequency hearing loss. **a** The three-cycle cisplatin regimen administered to CBA/CaJ mice. Each cycle consisted of 4 days of once-daily i.p. injections of cisplatin followed by 10 days of recovery. **b** Hearing loss following the cisplatin regimen as measured by auditory brainstem response recordings. Cisplatin caused a moderate to severe hearing loss across frequencies. $n = 22$ mice for cisplatin and saline groups. $n = 6$ mice for 60 days after cisplatin and saline groups. Statistical significance as determined by two-way ANOVA followed by the Holm–Šidák multiple comparisons test is displayed for cisplatin vs. saline. **c** Progression of hearing loss after each cycle of cisplatin and subsequent recovery. High-frequency hearing loss becomes evident after two cycles. Robust hearing loss across all frequencies is evident after three cycles. $n = 3–5$ for each cisplatin treatment time point, $n = 23$ for saline group. Two-way ANOVA followed by the Holm–Šidák multiple comparisons test was used to determine significance. **d** Percentages of surviving inner and outer hair cells (as compared to control averages) following the complete cisplatin regimen. Predominantly outer hair cells in the basal portion of the cochlea are lost. $n = 3$ mice in each group. **e** Distortion product otoacoustic emission (DPOAE) amplitudes (a measure of outer hair cell function) after each cycle and recovery period. Gray shaded area depicts the testing noise floor (the underlying background noise detected by the system microphone). $n = 3–5$ mice at each time point. Two-way ANOVA followed by the Holm–Šidák multiple comparisons test was used to determine significance. Data are expressed as mean ± s.e.m. ns not significant, $*P < 0.01$, $**P < 0.01$, $****P < 0.0001$

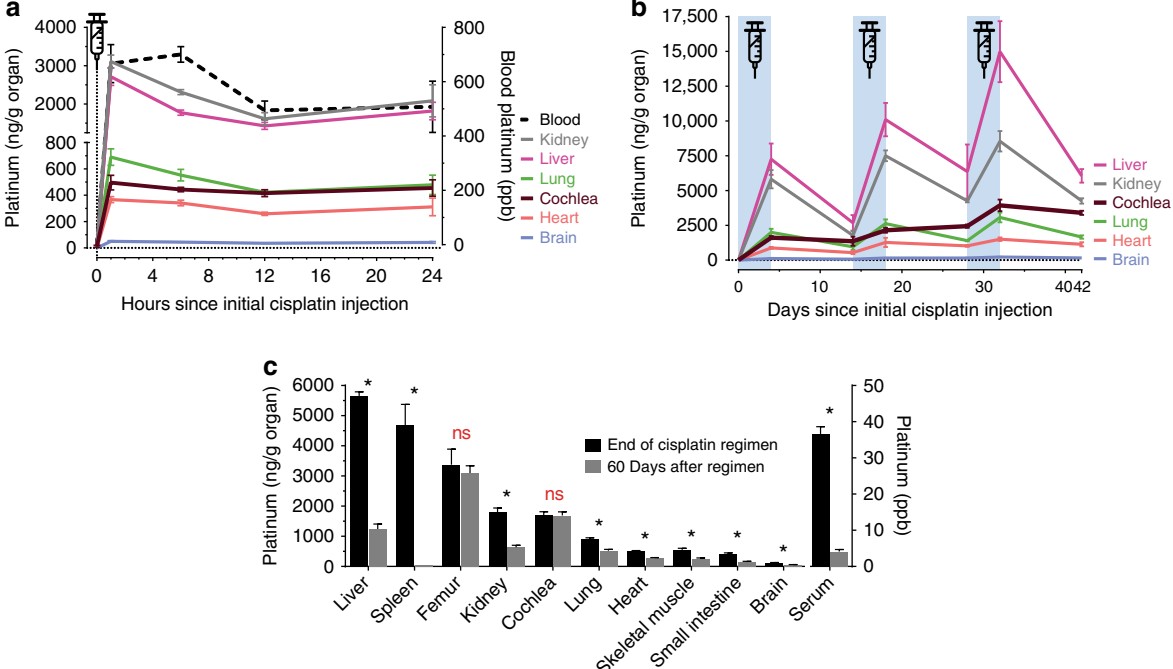

**Fig. 2** Cisplatin is readily cleared from most organs but is retained in the cochlea. **a** ICP-MS measured platinum concentrations in whole mouse organs isolated throughout the first 24 h following a single i.p. injection of cisplatin. 0-h data points represent mice prior to injection of cisplatin. Concentrations are normalized to organ mass (ng Pt per g of organ, left y-axis) or expressed as parts per billion (ppb, right y-axis) for whole blood. Platinum levels peak in most organs 1 h after injection. Concentrations vary significantly across organs and are highest in kidney and liver. $n = 3$–4 mice per time point. **b** Concentrations of cisplatin in whole mouse organs throughout the complete 42-day cisplatin regimen. The three, 4-day long cycles of once-daily i.p. cisplatin are indicated by blue shading. In most organs, platinum levels increase following injection and then decline during the recovery period. In contrast, platinum progressively accumulates in cochlea with no apparent elimination. $n = 3$–4 mice at each time point for all organs. **c** Concentrations of platinum in whole mouse organs at the end of the cisplatin regimen and following a subsequent 60-day recovery. During this recovery period, platinum was significantly eliminated from all organs except the cochlea and femur. $n = 3$–4 mice in each group. Multiple two-tailed t-tests with correction for multiple comparisons by the Holm–Šidák method were used to determine significance. Data are expressed as mean $\pm$ s.e.m. ns not significant, $^{*}P < 0.05$

relative to other cochlear regions. We correlate this accumulation with impairment of strial function. In human cochlear tissue, we employ laser ablation coupled to ICP-MS to directly visualize the distribution of cisplatin throughout the cochlea, again finding high accumulation in the stria vascularis. Our study highlights the importance of cisplatin pharmacokinetics in driving cisplatin ototoxicity and points to the stria vascularis as an important therapeutic target for prevention of cisplatin ototoxicity.

## Results

**A clinically relevant mouse model of cisplatin ototoxicity.** Previous studies of cisplatin ototoxicity in animals have utilized single high-dose systemic injection or direct cochlear administration, neither of which is reflective of clinical cisplatin administration[14–19]. We recently described a novel mouse model of cisplatin ototoxicity designed to be more analogous to clinical regimens[20]. In a refined version of this protocol, adult mice received three cycles of cisplatin with intervening 10-day recovery periods (Fig. 1a). Treated mice demonstrated significant hearing loss across all frequencies, most severe at the highest frequencies, similar to cisplatin ototoxicity in humans (Fig. 1b). Testing performed at the end of each cisplatin cycle and recovery period demonstrated that hearing loss progressed significantly with each subsequent cycle as well as during recoveries 1 and 3 (Fig. 1c). Hearing loss at the highest frequency (40 kHz) was significant after recovery 2, while changes in lower frequencies reached significance after cycle 3 (32, 22.4, and 8 kHz) or recovery 3 (16 and 11.2 kHz). Quantification of surviving mechanosensory hair cells in cisplatin-treated mice demonstrated near-total loss of

outer hair cells in the high-frequency basal cochlear region (Fig. 1d). Distortion product otoacoustic emissions (DPOAEs, a measure of cochlear outer hair cell function) showed significant, progressive deterioration with subsequent cycles of cisplatin and over the 60 days following the end of the regimen (Fig. 1e). DPOAE amplitudes were decreased even at low and middle frequencies, despite preservation of outer hair cells in the cochlear apical and middle regions that detect these frequencies, suggesting that the function of surviving outer hair cells was compromised. These results validate that our cisplatin regimen accurately models clinical cisplatin ototoxicity, and they suggest that hearing loss in mice progresses in the days and weeks following cessation of cisplatin administration, as has been reported clinically[21].

**Cisplatin is retained in the cochlea.** We first investigated cisplatin distribution immediately following a single injection. Peak levels of platinum were reached within 1 h across all mouse tissues studied, indicating rapid systemic distribution (Fig. 2a). Tissues from control (saline-injected) mice were studied in parallel, and these showed no appreciable platinum in this nor any other analyses (Supplementary Fig. 1). Sequential measurements throughout the 42-day cisplatin regimen revealed three general patterns of platinum tissue pharmacokinetics (Fig. 2b). In most organs, platinum levels peaked following each injection cycle and then troughed during the following recovery cycles. Platinum levels in brain remained very low, consistent with reports that cisplatin does not cross the blood–brain barrier[22,23]. The cochlea, however, exhibited unique pharmacokinetics among the organs analyzed. It accumulated platinum in a stepwise fashion with each

subsequent treatment cycle (Fig. 2b), indicating that in contrast to other organs, the cochlea has remarkably little capacity for eliminating cisplatin and its derivatives[24].

To examine the potential for long-term retention of cisplatin in the cochlea, we compared platinum levels at the end of the regimen (day 42) to those 60 days later (Fig. 2c). Most organs showed significant elimination of platinum during this period; the notable exceptions were cochlea and femur. These two organs showed no significant loss of platinum after 60 days. This long-term retention of platinum in the cochlea likely contributes to the unique susceptibility of the inner ear to cisplatin-induced damage.

**Cisplatin preferentially accumulates in stria vascularis.** Three cochlear regions have previously been implicated in cisplatin ototoxicity: the organ of Corti that contains the sensory hair cells and supporting cells, the spiral ganglion neurons that innervate the hair cells, and the stria vascularis that regulates the composition of the endolymph fluid[19]. ICP-MS analysis of microdissected samples from these regions (schematized in Fig. 3a) revealed rapid distribution of cisplatin species throughout all three regions within an hour of injection (Fig. 3b). Platinum accumulated in all three cochlear regions, with the highest levels in the stria vascularis by the end of the regimen (Fig. 3c). Sixty days later, platinum levels in serum had decreased by 89%, but

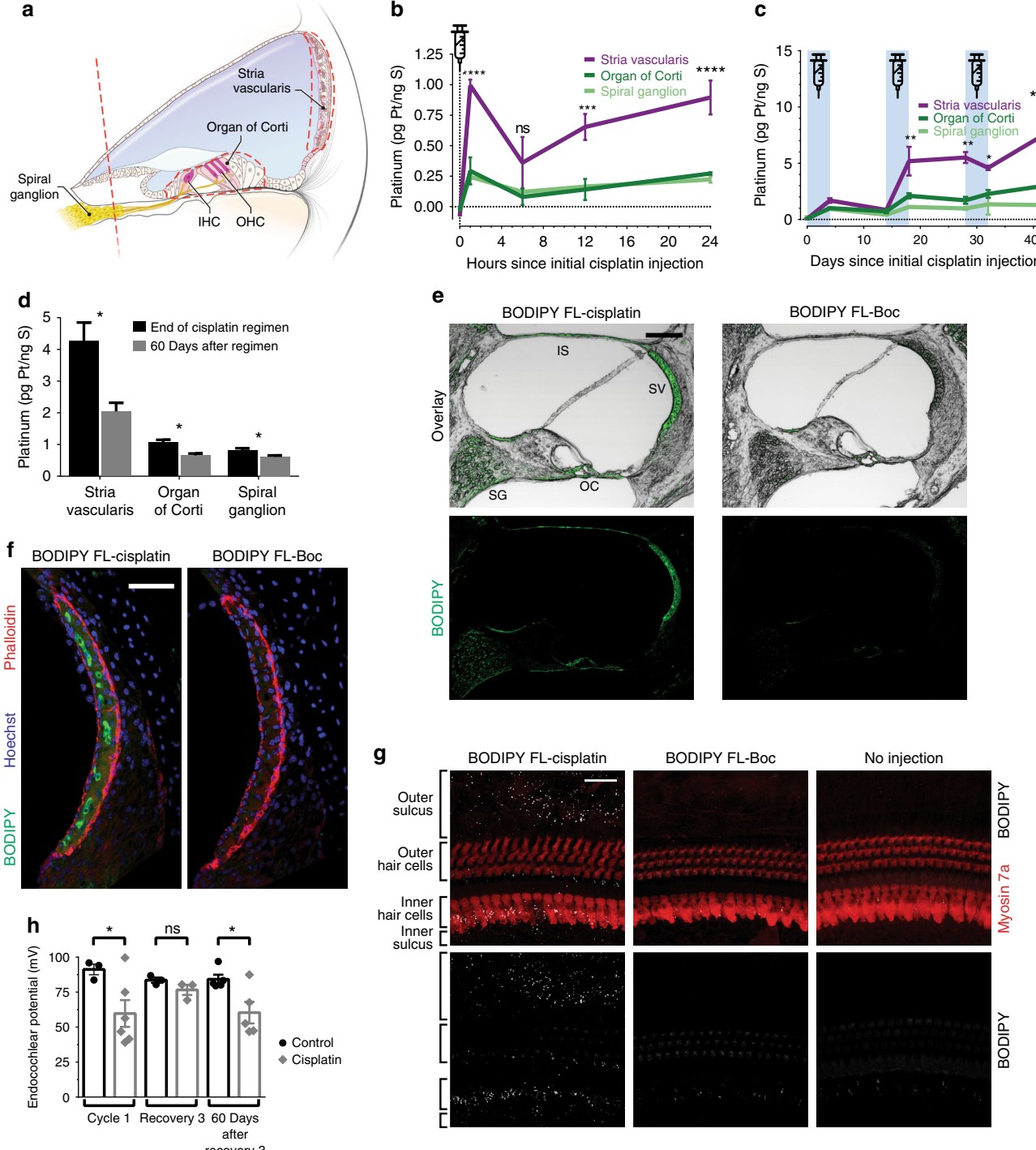

had only decreased by 52%, 39% and 25% in stria vascularis, organ of Corti, and spiral ganglion, respectively (Fig. 3d). Over this same time period, we did not detect any decrease in platinum levels in the whole cochlea (Fig. 2c). The remaining platinum in the cochlea may be retained in other regions, such as the cochlear bone, which represents a much larger percentage of the total cochlear mass than the regions we microdissected. Our data indicate that stria vascularis accumulates a higher concentration of cisplatin species than other regions of the cochlea.

We sought to visualize the cellular localization of cisplatin in cochlea through use of the fluorescent cisplatin conjugate BODIPY FL-cisplatin, recently shown to have cellular uptake similar to cisplatin[25,26] (Supplementary Fig. 2). Following a regimen of BODIPY FL-cisplatin that was dose-equivalent to that used for cisplatin, fluorescent signal filled the stria vascularis and could be faintly seen in the spiral ganglion, interscalar septum, and organ of Corti (Fig. 3e, f). No specific BODIPY fluorescence signal was apparent in cochlear tissue from control mice treated with the unconjugated (cisplatin-free) dye BODIPY FL-Boc, suggesting that the BODIPY ligand was not responsible for the uptake and retention of BODIPY FL-cisplatin that we observed. In organ of Corti whole mounts, no discernable fluorescent signal was present in outer or inner hair cells, though punctate fluorescence was seen in the cells of the inner and outer sulci (punctae were proximal to, but not within inner hair cells, Fig. 3g). These data agree with those obtained by ICP-MS analysis of cochlear microdissections in suggesting that the stria vascularis accumulates the highest levels of cisplatin, and that cisplatin accumulation in mechanosensory hair cells is more limited.

A primary function of the stria vascularis is to maintain the endocochlear potential[27], the electrochemical potential of endolymph that functions as the driving force for hair cell mechanoelectrical transduction. We assessed stria vascularis function by recording the endocochlear potential and found that cisplatin-treated mice had reduced endocochlear potentials both at the end of cisplatin cycle 1 and 60 days after the complete cisplatin regimen (Fig. 3h). Thus, the hearing loss and outer hair cell dysfunction measured in these mice can be at least partially explained by cisplatin-induced reduction in the endocochlear potential.

**Cisplatin is retained in the human cochlea indefinitely.** To investigate whether long-term cochlear accumulation of cisplatin species occurs in humans, we analyzed cochlear portions of postmortem human temporal bone tissue sections (Fig. 4b).

Sections from patients who had received cisplatin chemotherapy consistently contained significantly higher levels of platinum than sections from age- and sex-matched untreated patients. Platinum was retained in cochleae for at least 18 months after a patient's last cycle of cisplatin (Fig. 4a). Differences in the cisplatin regimens received by these patients precluded analysis of platinum levels over time. Cochlear sections from only one cisplatin-treated pediatric patient were available for analysis, and these contained significantly more platinum than any of the sections from adult patients ($P < 0.0001$), despite the pediatric patient having received a lower dose (total 180 mg/m$^2$ BSA) than most, if not all, of the adult patients (median total dose of 245 mg/m$^2$ BSA for all complete records). These results illustrate long-term retention of platinum in the human cochlea after cisplatin chemotherapy, which may be most pronounced in pediatric patients. These findings may explain both the reported progression of hearing loss for months after cessation of cisplatin chemotherapy and the increased susceptibility of pediatric patients to cisplatin ototoxicity[4,21,27–30].

To visualize the distribution of cisplatin within human cochlear sections, we employed laser ablation coupled to ICP-MS (LA ICP-MS). Across ablated sections from three patients, we observed platinum throughout all cochlear tissues with specific regions consistently containing higher concentrations (Fig. 4c). Platinum levels were consistently highest in the stria vascularis and in portions of the modiolus (the central axis of the cochlea), specifically at the boundary between the cochlear nerve and modiolar bone. High concentrations were also seen along the cochlear nerve fibers and along the endosteum lining the cochlear canal. Concentrations of platinum within the organ of Corti were indistinguishable from neighboring tissue, suggesting no specific accumulation of cisplatin within hair cells, although the resolution of the LA ICP-MS (25 μm) precludes absolute determination. Ablations from untreated patients displayed no platinum signal above background (Supplementary Fig. 3a). Laser ablation ICP-MS was also used to visualize cisplatin distribution within a mouse cochlear tissue section, and the observed pattern of cisplatin distribution was similar to that seen in human tissue, with high signal intensity in the stria vascularis and low signal intensity in the organ of Corti across cochlear turns (Supplemental Fig. 3b). Since the basal turn of the cochlea is more susceptible to cisplatin-induced damage than the apical turn, we analyzed the relative cisplatin signal intensity within the stria vascularis in each turn[19]. Cisplatin signal intensity was highest in the cochlear base, and it generally decreased with progression toward the apex (Supplemental Fig. 3c). Increased accumulation

**Fig. 3** The cochlear stria vascularis accumulates high levels of cisplatin, resulting in its dysfunction. **a** A cross-sectional illustration of cochlear architecture. The three regions of the cochlea: stria vascularis, organ of Corti and spiral ganglion, microdissected for ICP-MS analysis are indicated by dashed red lines. Endolymph fills the blue shaded region. IHC inner hair cell, OHC outer hair cell. **b** Platinum concentrations in microdissected cochlear regions in the first 24 h following a single i.p. injection of cisplatin. Cisplatin readily distributes throughout these cochlear regions within the first hour. Platinum concentrations are normalized to sulfur content, a surrogate for total tissue quantity[51]. $n = 3$–4 mice per time point. Statistical significance is displayed for stria vascularis values as compared to the other two cochlear regions. Two-way ANOVA followed by the Holm–Šidák multiple comparisons test was used to determine significance. **c** Concentrations of platinum in cochlear regions throughout the complete cisplatin regimen. Platinum progressively accumulates in all three regions, but most significantly in stria vascularis. $n = 3$ mice per time point. Statistical significance is displayed for stria vascularis values as compared to other two regions. Two-way ANOVA followed by the Holm–Šidák multiple comparisons test was used to determine significance. **d** Concentrations of platinum in cochlear regions at the end of the cisplatin regimen and following a subsequent 60-day recovery. All regions show a decrease in platinum concentrations. $n = 4$ mice per group. Unpaired two-tailed $t$-tests were used to determine significance. **e** Representative low-magnification confocal slices from cochlear cryosections of mice treated with a full regimen of BODIPY FL-cisplatin or the control (no cisplatin) conjugate BODIPY FL-Boc. SG spiral ganglion, OC organ of Corti, SV stria vascularis, IS interscalar septum. Scale bar = 100 μm. **f** Representative maximum intensity projections of thin confocal stacks of the stria vascularis only. Scale bar = 50 μm. **g** Representative maximum intensity projections of confocal stacks of organ of Corti whole mounts. Scale bar = 25 μm. **h** Endocochlear potentials (EPs) measured in the endolymph following cisplatin treatment. EPs decline at 24 h after cisplatin but recover somewhat by the end of the regimen. However, 60 days after cessation of cisplatin administration, EPs are still significantly reduced. $n = 3$–6 mice per group; data points represent individual mice. Unpaired two-tailed $t$-tests were used to determine significance. Data are expressed as mean ± s.e.m. ns not significant, *$P < 0.05$, **$P < 0.01$, ***$P < 0.001$, ****$P < 0.0001$

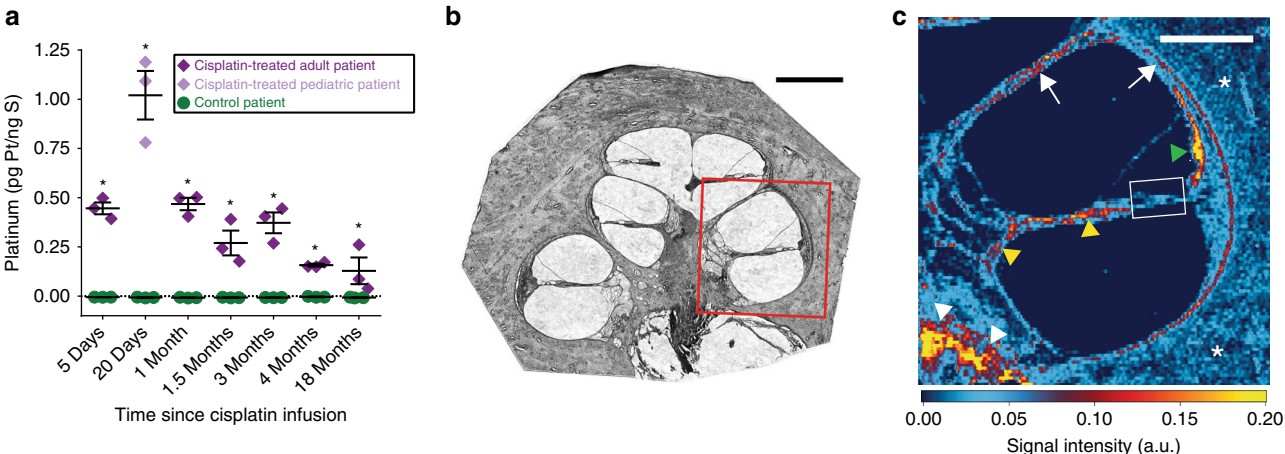

**Fig. 4** Cisplatin is retained in human cochleae long after cessation of treatment. **a** Platinum concentrations measured in cochlear sections of human temporal bones. Matched pairs of patients who received cisplatin chemotherapy and control (cisplatin-naive) patients are plotted by the time elapsed between the patient's last cisplatin infusion and their death. Each data point represents an individual 20 μm cochlear section. Multiple two-tailed *t*-tests with correction for multiple comparisons by the Holm–Šidák method were used to determine significance. Data are expressed as mean ± s.e.m. *$P < 0.05$. **b** Representative brightfield image of a cochlear section prior to ICP-MS analysis. Similar sections were digested whole and analyzed for total cisplatin content. This section was laser ablated in the region outlined in red. Scale bar = 1 mm. **c** Laser ablation ICP-MS image of platinum distribution in the area in the red box in **b** from a patient who was treated with cisplatin. This patient died 25 days after their last cisplatin infusion. Green arrowhead marks stria vascularis, yellow arrowheads mark cochlear nerve fibers, white arrowheads mark the boundary between the cochlear nerve and the bone of the cochlear modiolus, white arrows mark the endosteum which lines the cochlear canal, asterisks mark surrounding cochlear bone, and white box frames organ of Corti. Scale bar = 500 μm. Platinum signal intensity values are in arbitrary units

of cisplatin in the cochlear base is consistent with the increased susceptibility of that region to cisplatin-induced damage. Together, these laser ablation ICP-MS results demonstrate prominent retention of cisplatin species in the stria vascularis of the human inner ear, and they suggest that cisplatin retention may be highest in the cochlear base.

## Discussion

Ototoxicity is a serious adverse effect of cisplatin chemotherapy, experienced by patients receiving either high or low-dose regimens[7,31,32]. Here we show that platinum is retained in the cochleae of experimental mice and treated cancer patients months-to-years after cisplatin chemotherapy. Though previous work has focused upon identifying the pathways responsible for cellular hypersensitivity to cisplatin within the cochlea, our results suggest that it is hyper-accumulation and not hypersensitivity that drives cisplatin ototoxicity. Cell types within the cochlea likely experience higher cumulative drug exposure than cell types in less susceptible tissues. This is in line with previous evidence for a correlation between the concentration of different platinum chemotherapeutics in cochlear endolymph and their level of ototoxicity[33].

Long-term retention of cisplatin in the cochlea may explain the delayed progression of cisplatin-induced hearing loss that we observed in our mouse model and has also been reported clinically[21]. These findings suggest that hearing testing should be consistently included as a component of long-term follow-up care after cisplatin chemotherapy in order to identify all patients who could benefit from hearing rehabilitation. Patients receiving only short-term audiometric testing could suffer from undiagnosed, and thus untreated, late progression of hearing loss. This audiometric monitoring should also include testing at frequencies up to 16 kHz, as extended range audiometry has been shown to improve the sensitivity for detecting cisplatin ototoxicity[34].

Pediatric patients are most susceptible to cisplatin ototoxicity[4,21,27–30]. In our study, those cochlear tissue sections from the only pediatric patient donor available contained higher

concentrations of platinum than those from adult counterparts, despite the patient receiving a lower total dose (per body surface area) of cisplatin. This finding suggests that differences in cisplatin pharmacokinetics between adult and pediatric patients may explain the increased susceptibility of children to cisplatin-induced hearing loss.

Outside the cochlea, we also found long-term retention of prominent levels of cisplatin in the long bones (femur). Long-term exposure to circulating platinum is associated with several late toxicities of cisplatin treatment, and platinum has been detected in the blood of chemotherapy patients more than 20 years after treatment[34–39]. Such long-term elevated platinum levels in the blood are presumed to be a result of cisplatin species being re-mobilized from an unidentified tissue reservoir. Our results suggest that bone may serve as a reservoir for platinum, and prolonged release of platinum from bone may mediate the late toxicities of cisplatin in cancer survivors. In support of this idea, cisplatin has recently been demonstrated to bind extensively to, and slowly dissociate from, type I collagen, the major protein component of bone[40]. Our results suggest that platinum may have pharmacokinetics akin to that of lead, another heavy metal. In cases of toxicity, lead is known to distribute into bone, where it has a half-life of years-to-decades, and from which it can be exchanged back into the blood[41]. Lead toxicity is similarly associated with hearing loss[42]. Our ICP-MS-based approach allowed for equal detection of bound and free (unbound) platinum, but not differentiation of the two. Further study of cisplatin reactivity with proteins and other biomolecules may further elucidate the exact mechanisms by which this drug causes cytotoxicity both within and outside the cochlea.

Our data provide the first map of platinum distribution in the human cochlea after cisplatin chemotherapy, and they demonstrate long-term retention of cisplatin in the inner ear. Our results using multiple techniques point to the stria vascularis as the region of the cochlea exposed to the highest levels of platinum, and we find that this exposure results in impairment of strial function. Efforts toward prevention of cisplatin ototoxicity have focused almost exclusively upon protection of the sensory hair

cells[43,44]. We find no discernable accumulation of the BODIPY FL-cisplatin conjugate within inner or outer hair cells, and laser ablation ICP-MS showed no increased platinum signal from the organ of Corti, suggesting low accumulation of cisplatin in cochlear hair cells. Hair cell death may be secondary to impaired maintenance of the endolymph by the stria vascularis, which is known to cause hair cell death, and/or hair cells may be more sensitive to cisplatin than other cochlear cell types[45]. Regardless, our findings suggest that we must look beyond hair cells to overcome cisplatin ototoxicity. As the vascular tissue of the cochlea, the stria vascularis also represents a likely entry point for cisplatin[46]. Our results point to a strategy aimed at prevention of cisplatin uptake into the stria vascularis as a promising therapeutic approach to prevention of cisplatin ototoxicity. The mouse model and ICP-MS techniques detailed here will allow for the screening of bioactive molecules that may reduce the uptake of cisplatin into the cochlea. Local administration of such a drug could prevent ototoxicity without interfering with the anti-cancer efficacy of cisplatin.

## Methods

**Animals**. One hundred and four CBA/CaJ male and female mice (The Jackson Laboratory), age 6–16 weeks, were used in this study. All animal procedures were approved by the NIDCD/NINDS Animal Care and Use Committee (protocol #1327). Mice were assigned numbers sequentially and then a random number generator was used to partition them into the cisplatin-treated experimental group or saline-treated control group. Male and female mice were randomized independently to ensure even sex distribution throughout experimental groups. Of the 104 animals, 90 ($n = 48$ Cisplatin-treated, $n = 42$ Control) underwent auditory testing prior to and at the end of treatment; 25 were used for additional measurements of endocochlear potential (EP). Cisplatin-treated (APP Pharmaceuticals) mice received three rounds of once-daily i.p. injections (3.5 mg/kg) for 4 days followed by 10 days of recovery (total 42 days). Saline-treated mice received a parallel regimen of saline given at the equivalent volume as their cisplatin-treated counterparts. The investigator was not blinded to the treatment group. Daily subcutaneous injections (up to 2 ml) of 0.9% NaCl and Plasma-LYTE (Abbott Laboratories) were provided in addition to soft pellet chow and fresh fruit to mice in need of supplemental nutrition. Some cisplatin-treated mice required hand feeding of a high-calorie liquid supplement (STAT®, PRN Pharmacal) to maintain acceptable body weight. For some experiments, subsets of mice were assessed and euthanized for tissue examination after each cycle of treatment. Of those mice that received the entire cisplatin regimen, 100% (27) survived the regimen, and 85.2% (23) survived through the subsequent auditory testing.

The remaining 14 mice were used for in vivo BODIPY experiments. BODIPY FL-cisplatin, a conjugate of cisplatin and the low molecular weight fluorophore BODIPY FL, and the free-dye (no cisplatin) compound BODIPY FL-Boc (Supplementary Fig. 2) were synthesized by the Imaging Probe Development Center of the National Heart, Lung, and Blood Institute Intramural Program. Mice in the experimental group ($n = 7$) received a regimen of BODIPY FL-cisplatin while control mice ($n = 7$) received BODIPY FL-Boc. The dosing regimen entailed three cycles, each consisting of 4 days of once-daily i.p. injections. In order to deliver the molar equivalent dose as used in the cisplatin regimen, BODIPY FL-cisplatin injections were dosed at 7.3 mg/kg of mouse mass, and BODIPY FL-Boc injections were dosed at 5.1 mg/kg. Both compounds were delivered in a total of 1 mL of saline-based solution containing 61.6 mg/kg of methyl-β-cyclodextrin (Sigma) and 2 mL/kg of dimethylformamide (Alfa Aesar). For $n = 3$ mice in each experimental group, cochleae were collected on the day following completion of the regimen.

**Auditory testing**. Mice were anesthetized with ketamine (Putney Inc., Portland, ME; 100 mg/kg i.p.) and xylazine (Akorn Inc., Lake Forest, IL; 10 mg/kg i.p.). Supplemental injections at 1/3-1/2 of the original dose were administered if needed. Animals were placed on a temperature-controlled (37 °C) heating pad (World Precision Instruments T-2000, Sarasota, FL, USA) inside a sound isolation booth (Acoustic Systems, Austin, TX, USA). Distortion-product otoacoustic emissions (DPOAEs) and auditory brainstem responses (ABRs) were recorded using Tucker-Davis Technologies (TDT; Alachua, FL, USA) hardware (RZ6 Processor) and software (BioSigRZ).

The DPOAE at $2f1-f2$ was recorded in the mouse ear canal using an ER-10B + microphone (Etymotic, Elk Grove Village, IL, USA) covered with a modified pipette tip. The DPOAE primary tones were presented at constant levels of $f1 = 65$ dB SPL and $f2 = 55$ dB SPL at 14 $f2$ frequencies between 8000–40000 Hz ($f2/f1 = 1.25$) via two TDT MF-1 speakers. Mean noise floors were calculated from six spectra surrounding the DPOAE frequency (+/− 150 Hz).

ABRs were generated using Blackman-gated tone burst stimuli (3 msec, 29.9/sec, alternating polarity) at 8, 11.2, 16, 22.4, 32 and 40 kHz via a closed-field TDT MF-1 speaker. Subdermal needle recording electrodes (Rhythmlink, Columbia, SC) were placed at the vertex, under the test ear, and at the base of the tail (ground). Average waveforms from 1024 presentations were generated, amplified (20x), filtered (.3–3 kHz), digitized (25 kHz) and stored for offline analysis. For each test frequency, recording began at 80 dB SPL and decreased until the ABR waveform was no longer evident. If no response was obtained at 80 dB SPL, testing was performed at a maximum level of 90 dB SPL. ABR thresholds were determined by visual inspection of stacked waveforms for the lowest stimulus level that yielded repeatable waves. Threshold determinations were verified by a second investigator who was blinded to the treatment group.

**Endocochlear potential recordings**. Endocochlear potentials (EPs) were recorded from mice under deep anesthesia (Avertin, 15.1 mg/ml at 0.35 mg per kg body weight). Mice were mounted in the supine position in a secure head holder, and a midline incision in the neck and muscles surrounding the trachea was made in order to place a tracheal cannula between the 4th and 5th ring. Careful to prevent excessive bleeding, muscles underlying the thyroid were separated to expose the bulla. An initial opening in the bulla was made using a #11 blade and expanded with micro forceps to reveal the round window membrane while ensuring no contact with the stapedial artery. The darkened region near the basilar membrane where hair cells are located in the base of the cochlea was identified and used as a target for electrode placement.

A 1 M KCl gel (agarose) filled glass pulled pipette reference electrode with Ag-AgCl wire (Worcester Polytechnic Institute, 1B-150F-4) was placed subcutaneously in the abdomen. Using a micromanipulator (Narishige, US-3f), the tip of a small 1 M KCL filled glass pulled recording pipette with a Ag-AgCl wire electrode (Worcester Polytechnic Institute, 1B-100F-4) was advanced through the round window membrane into the scala tympani and then through the basilar membrane into the scala media, where an EP was recorded. Signals were digitally recorded (Digidata 1440 A and AxoScope 10.3; Axon Instruments). An EP was confirmed as a >10 mV potential that was maintained for 100 s and returned to 0 mV as the electrode retracted to its original position in scala tympani. At the end of the procedure, mice were euthanized without recovery from anesthesia.

**Tissue collection for ICP-MS**. Mice were euthanized by $CO_2$ inhalation followed by decapitation. To obtain serum samples, whole blood was drawn from bilateral external jugular veins, allowed to clot for at least 20 min at room temperature, and then spun at 2000 g for 15 min at 4 °C, with only the supernatant being saved. Whole organs were dissected out, rinsed in ultrapure water, and patted dry. To obtain cochlea, whole inner ears were removed, and remaining portions of the boney auditory bulla as well as the vestibular organs were dissected away. Femurs were flushed of bone marrow using ultrapure water through a 25-gauge needle. All samples were stored at −80 °C until analysis.

**Cochlear microdissection and cryosectioning**. Cochleae used for microdissection or cryosectioning were fixed in 4% paraformaldehyde overnight at 4 °C and were washed in PBS before decalcification in 0.5 M EDTA solution at pH 8.0 and room temperature for 48 h. Decalcified cochleae were microdissected using ophthalmic straight knives (Accutome) and Dumont #55 Forceps (Fine Science Tools). Stria vascularis was peeled from the cochlear lateral wall; organ of Corti and spiral ganglion samples were cut from their surrounding tissues. The entirety of each region from cochlear apex to base was isolated and analyzed as one.

Decalcified cochleae to be cryosectioned were first incubated overnight in 30% sucrose in PBS at 4 °C with gentle rotation, then incubated in a 1:1 solution of 30% sucrose and Tissue-Tek OCT compound (Sakura) for 4 h at 4 °C, and finally incubated in OCT overnight at 4 °C. The following day, cochleae were oriented within OCT-containing cryomolds (Tissue-Tek, Intermediate; Sakura) and frozen on a cooling bath of absolute ethanol and dry ice. Frozen sections of 12 μm thickness were cut on a cryostat (Leica CM), captured on glass slides (Superfrost Plus; Thermo Scientific), and allowed to air dry for 1 to 2 h at room temperature.

**Immunohistochemistry**. Microdissected or sectioned tissue was incubated for 2 h at room temperature in blocking solution composed of 0.5% triton X-100 (Sigma-Aldrich), 2% BSA (Sigma-Aldrich), and 0.8% normal goat serum in PBS. Tissues were stained overnight at 4 °C with primary antibody against Myosin VIIa (Product MYO7A 138-1 deposited to the Developmental Studies Hybridoma Bank by Orten, D.J.), diluted 1:100 in blocking solution, followed by Alexa Fluor 546 conjugated goat anti-mouse IgG (Invitrogen, catalog #A11003, 2 mg/ml stock diluted 1:500 in blocking solution) for 4 h at room temperature. F-actin was stained with Alexa Fluor 647 Phalloidin (Invitrogen, catalog #A31573, 2 mg/ml stock diluted 1:60 in blocking solution), and nuclei were stained with Hoechst 33342 (Molecular Probes, catalog #H3570, 10 mg/ml stock diluted 1:500 in blocking solution). Tissues were mounted in Fluoromount-G (Southern Biotech) and imaged on an LSM 780 laser scanning confocal microscope (Carl Zeiss Microscopy). Experimental and control samples were imaged under identical acquisition settings, including laser line attenuation and detector gain. Post-acquisition processing was performed in ImageJ2[47] and was applied equally and across the entire image to all images used for comparison. To quantify hair cell survival, Myosin

VIIa positive cells in 135-µm-long regions from the center of each cochlear turn (apex, middle, and base) were manually counted.

**Patient cochlear sections**. The NIH Office of Human Subjects Research Protections determined that the study to be exempt from Institutional Review Board review as it utilized existing records and post-mortem pathological specimens. Samples were obtained via a tissue transfer agreement with the NIDCD National Temporal Bone, Hearing and Balance Pathology Resource Registry at the Massachusetts Eye and Ear Infirmary. The National Temporal Bone Registry online database was queried to identify banked temporal bone samples from patients who had received a chemotherapy regimen including cisplatin. Patients with noted anatomical defects and/or tumor invasion involving—or in close proximity to—the cochlea were excluded. Eight suitable cisplatin-treated patients were identified. The database was then queried to identify a sex- and age-matched control patient, who had never received cisplatin, to match each cisplatin-treated patient. Three to four approximately mid-modiolar 20 µm sections were prepared from each patient. Celloidin embedding medium was removed using sodium hydroxide and methanol according to a published protocol[48]. The final rinse was performed in ultrapure water. Sections were manually trimmed to only include the cochlea and immediately surrounding bone, and were then tile-scan imaged at ×10 under brightfield optics (Zeiss LSM 780). Images were shading corrected and stitched in Zen 2.1 software (Carl Zeiss Microscopy). Sections were then either digested and analyzed for total platinum content, or were analyzed via laser ablation ICP-MS.

**Inductively coupled plasma mass spectrometry**. Samples for analysis by inductively coupled plasma mass spectrometry were labeled only by a unique identifying number, and so the investigators performing the analyses were blinded to the treatment group. Platinum measurements were performed using a NexION 350D inductively coupled plasma-mass spectrometer (Perkin Elmer). The instrument was tuned daily for optimum performance and sensitivity. All whole organs were weighed prior to analysis. Organs were digested first with trace metal nitric acid (Fisher Chemical) and incubated for 20 min at 65 °C. An equal volume of hydrogen peroxide (Optima Grade, Fisher Chemical) was added and the incubation was repeated. Samples were diluted 1:100 with 2% nitric acid prior to running. A quantitative analysis method in standard mode measuring the platinum 195 isotope was used to determine platinum concentrations. A platinum standard (SPEX CertiPrep) curve was generated with 0.2, 1.0, and 2.0 ppb standards. Iridium isotope 193 was used as the internal standard (10 ppb stock concentration, SPEX CertiPrep) and was added inline using the micropPREP FAST SV500 injection system (Elemental Scientific, Inc.). Each sample was run in duplicate and platinum concentrations were normalized to organ weight.

Mouse cochlear microdissection samples were prepared by removing residual liquid by speed vacuum concentration. Samples were digested as described above, but volumes of 10 µl of nitric acid and 10 µl of hydrogen peroxide were used. Samples were resuspended to a final volume of 500 µl with 2% nitric acid. Platinum 195 and iridium 193 isotopes were measured as described above. Phosphorous and sulfur (Inorganic Ventures) were additionally measured in dynamic reaction mode (DRC) using oxygen at a 1.2 ml/min gas flow rate. The isotopes PO 47 and SO 48 were measured using a 50, 250, and 500 ppb standard curve. Beryllium isotope 7 was used as the internal standard (500 ppb stock concentration, SPEX CertiPrep) and added inline. Platinum values were normalized to sulfur values for each microdissected sample since the cochlear microdissected samples were too small to accurately use sample weight for normalization.

Human temporal bone specimens were removed from the glass slide under water using a new scalpel and quantitatively transferred to a 1.5 ml tube. Each sample was digested with 50 µl nitric acid and 50 µl hydrogen peroxide for 2 h at 65 °C. Each sample was re-suspended to a final volume of 500 µl with 2% nitric acid. Platinum, sulfur, and phosphorous were measured by ICP-MS and normalized as described above.

**Laser ablation ICP-MS**. Laser ablation ICP-MS was utilized to localize cisplatin in human temporal bone samples using a NWR 213 solid state laser ablation system (Electro Scientific Industries, Inc., Portland, OR). Conditions for the laser were as follows: 20 Hz repetition rate in continuous mode using 20% energy. A 25 × 25 µm spot size was generated using a 20 µm line spacing in Y. The scan speed was 50 µm/sec and 650 ml/min helium gas flow. Platinum 195 was analyzed with calcium 43 serving as a control. Heat maps were generated using Iolite software (https://iolite-software.com/)[49]. Platinum heat maps from experimental and control samples were scaled equally for intensity.

An additional laser ablation ICP-MS image of a cochlear section from a cisplatin-treated mouse was collected using a NWR193 laser ablation system equipped with a Bloodhound sample chamber for rapid particle transport (Electro Scientific Industries, Inc. Portland, OR), connected to an Agilent 7700 ICP-MS (Agilent Technologies, Santa Clara, CA). The laser parameters were optimized to minimize experiment duration while maintaining sensitivity of 195Pt above background: 5 × 5 µm spot size, 250 µm scan speed, 250 Hz rep rate, line spacing of 5 µm, and fluence of 6 J/cm$^2$. Helium gas flow of 800 ml/min delivered ablated particles to the ICP-MS, which monitored Platinum 195 using a duty cycle of 0.0198 s. Images were generated and interrogated using Iolite software v3.6.

**Statistics**. Data are presented as mean ± s.e.m. throughout. Each replicate was derived from an individual animal for all mouse experiments. Replicates in Fig. 4a represent individual cochlear tissue sections. Data graphing and all statistical analyses were performed using GraphPad Prism 6 software. No statistical methods were used to predetermine sample sizes, but sample sizes were in line with those necessary for detection of an effect in our previously reported in vivo studies[20,50]. Statistical significance was determined by two-way ANOVA followed by the Holm–Šidák multiple comparisons test or by unpaired two-tailed t-tests with or without correction for multiple comparisons by the Holm–Šidák method. Specific statistical tests are noted in respective figure legends. Data normality was tested by the D'Agostino–Pearson omnibus K2 test when n was sufficient to do so. Data presented in Supplementary Fig. 3c failed the normality test and were therefore analyzed by the Kruskal–Wallis test. Alpha was set at 0.05 in all cases.

**Data availability**. All data supporting the findings of this study are available within the article and its Supplementary Information files or from the corresponding author upon request.

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

## Acknowledgements

We are grateful to the Imaging Probe Development Center of the National Heart, Lung, and Blood Institute for synthesizing the BODIPY compounds and director R. Swenson for technical guidance on their in vivo administration; J. O'Malley (Massachusetts Eye and Ear) for preparing the temporal bone sections; A. Hoofring for illustrating cochlear anatomy; T. Fitzgerald of the NIDCD mouse auditory testing core (project number ZIC DC000080) for training in audiometric techniques; Perkin Elmer for their assistance with laser ablation analysis; the animal care staff of the John Edward Porter Neuroscience Research Center for their attentive care of cisplatin-treated mice; T. Friedman and M. Poirier for critiquing the manuscript; and the donors to National Temporal Bone Registry for their generous anatomical gifts. This research was funded by the Intramural Programs of the National Institute on Deafness and Other Communication Disorders (project number 1ZIADC000079) and the National Institute on Minority Health and Health Disparities.

## Author contributions

A.M.B. designed and coordinated the experiments, performed the experiments, interpreted the data, and wrote the manuscript. A.E.R. designed the experiments, performed the experiments, and interpreted the data. E.D.S. performed the ICP-MS measurements and analyzed the resulting data. K.A.F. and K.K.S. treated the mice and performed auditory testing. K.M.M. analyzed the data from the laser ablation ICP-MS experiments. M.D.H. conceived the study and designed the experiments. L.A. conceived and directed the study and performed the ICP-MS measurements and data analysis. L.L.C. conceived and directed the study and co-wrote the manuscript. All authors edited the manuscript.

## Additional information

**Competing interests:** K.M.M. is an Applications Specialist with Electro Scientific Industries, Inc., which manufactures the laser ablation modules utilized in this study. The remaining authors declare no competing financial interests.

