## [Peer Review File · Nature Communications]

Reviewers' comments:

Reviewer #1 (Remarks to the Author):

This manuscript provides some novel information concerning the prolonged retention of cisplatin in the mouse and human cochlea. The investigators show a stepwise accumulation of cisplatin in the cochlea over three cycles of treatment and the levels remain relatively unchanged 60 days following the last treatment cycle. This finding suggests that the residual cisplatin in the cochlea could still provide hearing loss long after cancer chemotherapy is terminated. This would suggest that patients should be tested for hearing loss months to years following cisplatin chemotherapy. The authors also suggest that the unique pharmacokinetics of cisplatin in the cochlea could explain the heightened sensitivity of this organ to cisplatin toxicity. The current data would support such a conclusion. The impact of the data is enhanced by findings from human temporal bone samples showing long-term retention of cisplatin in the cochlea following cisplatin chemotherapy. I have several points of clarification and critiques which are detailed below.

1. It is not clear what the survival rate of mice undergoing cisplatin treatment protocol is. Generally some attrition (~10-25%) is seen with the single dose protocol.
2. What accounts for the killing of outer hair cells and loss of DPOAE amplitudes if these cells are not accumulating much cisplatin. In fact the toxicity of the stria vascularis appears lower and less progressive over the treatment duration and 60 days following termination of drug treatment.
3. Are the samples analyzed for the regional distribution of cisplatin in the cochlea (Fig. 3d) obtained from the whole length of the cochlea or taken from specific regions? Is there a base to apex difference in distribution of cisplatin in the cochlea? What are the levels of cisplatin in the perilymph and endolymph?
4. Comment: Fig 1. Hearing loss (ABRs) appears greater in Recovery 1 (low frequency) than in recovery 2. This is not seen in DPOAE amplitudes.
5. There appears to be temporal differences in loss of ABRs, DPOAE amplitudes and EP despite greater accumulation of cisplatin in the cochlea (especially stria vascularis) over the course of treatment. Any explanation?
6. The molecular weight of BODIPY FL cisplatin is about twice as that of cisplatin. I wonder if this difference could partly explain its differential uptake in the organ of Corti versus stria vascularis via different transporters and/or ion channels.
7. It is not clear whether the cisplatin detected by ICP-MS is active, could be readily activated or is sequestered by being covalently incorporated into antioxidant molecules, proteins or DNA. Thus, it is not clear whether this "retained" cisplatin could provide "active" cisplatin at levels high enough to produce subsequent (and progressive) hearing deficits.

Reviewer #2 (Remarks to the Author):

This work is a new approach to understanding cisplatin ototoxicity and is convincing. The major finding is the mapping of cisplatin distribution within the human cochlea for the first time. The results indicate that cisplatin readily enters the inner ear but is very poorly cleared from it.

However I cannot see the paper as being of interest to a wider field - it is specialized. Further, the techniques themselves are not novel - they are simply analytical applied to this system. Whether these findings will change thinking in the field is doubtful at this point - perhaps if a more substantive approach were taken to answering the point the authors make -

"Our findings point to a strategy aimed at prevention of cisplatin uptake into the stria vascularis as a promising therapeutic approach to prevention of cisplatin ototoxicity." - the paper would be strengthened and perhaps clinical practitioners would be excited.

Another concern I have is that my understanding is that ototoxicity is a consequence of high-dose cisplatin. While I appreciate the technical difficulties and time involved it would be of interest to examine dose-dependence.

Overall I cannot recommend publication.

Reviewer #1 (Remarks to the Author):

1. It is not clear what the survival rate of mice undergoing cisplatin treatment protocol is. Generally, some attrition (~10-25%) is seen with the single dose protocol.

Our objective while developing our cisplatin ototoxicity protocol was to consistently induce significant hearing loss, but maintain a high survival rate. Of the 27 mice which received the full three cycles of cisplatin for this study, 27 (100%) survived the regimen and 23 (85.2%) survived the final auditory testing (which is performed under general anesthesia and is often when frail mice will perish). This survival rate is in line with the overall rate across many cisplatin-treated groups of mice in our laboratory. We find survival rates vary somewhat with the scientist directly caring for the mice, and experienced lab members have achieved survival rates as high as 100%.

In response to the Reviewer's comment, the survival rate information has been added to the Methods (lines 232-234): "Of those mice that received the entire cisplatin regimen, 100% (27) survived the regimen, and 85.2% (23) survived through the subsequent auditory testing."

2. What accounts for the killing of outer hair cells and loss of DPOAE amplitudes if these cells are not accumulating much cisplatin. In fact, the toxicity of the stria vascularis appears lower and less progressive over the treatment duration and 60 days following termination of drug treatment.

A portion of the loss in DPOAE amplitude is likely a result of stria vascularis dysfunction and not entirely a result of hair cell loss. Our data demonstrate a partial discord between the numbers of surviving outer hair cells and the DPOAE amplitudes. We observe minimal loss of outer hair cells in the apical and middle cochlear turns, yet DPOAE amplitudes are decreased in low and middle frequencies. Proper outer hair cell function, and thus the DPOAE amplitude, are dependent upon the endocochlear potential, which we show to be decreased. Of course, we do also note significant loss of outer hair cells. This too may be at least partially explained by strial dysfunction. As hair cells are bathed by endolymph, the composition of which is regulated by the stria, strial dysfunction may result in an altered hair cell extracellular environment and has been shown to result in loss of hair cells (Liu et al., 2016). Hair cells may also be more sensitive to cisplatin than other cochlear cell types. Previous work by other groups has implicated accumulation of reactive oxygen species, BAK1 and BAX signaling, and p53 in the cisplatin toxicity to hair cells (Callejo, Sedó-Cabezón, Juan, & Llorens, 2015).

In response to the Reviewer's comment, the following has been added to the manuscript:

Results (lines 76-78): "DPOAE amplitudes were decreased even at low and middle frequencies, despite preservation of outer hair cells in the cochlear apical and middle regions that detect these frequencies, suggesting that the function of surviving outer hair cells was compromised."

Discussion (lines 210-214): "We find no discernable accumulation of the BODIPY FL-cisplatin conjugate within inner or outer hair cells, and laser ablation ICP-MS showed no increased platinum signal from the organ of Corti, suggesting low accumulation of cisplatin in cochlear hair cells. Hair cell death may be secondary to impaired maintenance of the endolymph by the stria vascularis, which is known to cause hair cell death, and/or hair cells may be more sensitive to cisplatin than other cochlear cell types (Liu et al., 2016)."

3. Are the samples analyzed for the regional distribution of cisplatin in the cochlea (Fig. 3d) obtained from the whole length of the cochlea or taken from specific regions? Is there a base to apex difference in distribution of cisplatin in the cochlea? What are the levels of cisplatin in the perilymph and endolymph?

For the data presented in Figure 3d, samples were taken from the entire cochlear length. We performed the analysis this way because of the challengingly small amount of tissue available from each region (stria vascularis, organ of Corti, and spiral ganglion) in a single mouse cochlea. Therefore, we do not have information regarding the base-to-apex distribution of cisplatin for that analysis. However, we agree that this is an interesting question given the well-described differences in severity of cisplatin-induced damage occurring between the base and apex. Thus, in response to the Reviewer's comment we have now performed an additional analysis which we report in the revised manuscript. Using a high-resolution laser ablation ICP-MS system (to which we have very limited access), we analyzed cochlear tissue from a cisplatin-treated mouse and measured relative cisplatin signal intensity in the stria vascularis in each cochlear turn (Supplementary Figure 3b-c). We find cisplatin signal intensity to be highest in the cochlear base and to decrease towards the apex. This finding is consistent with the existing knowledge that cisplatin ototoxicity progresses from base to apex (high frequencies to low frequencies). We present these findings in Supplemental Figure 3b & 3c in the revised manuscript. We do not have the technical capability to measure cisplatin in the perilymph and endolymph for the mouse, but this has previously been investigated in the guinea pig by other research groups (Hellberg et al., 2009; Laurell, Andersson, Engstrom, & Ehrsson, 1995)

In response to the Reviewer's comment, the following has been added to the manuscript:

Supplemental Figure 3 has additional panels b and c

Results (lines 158-166): "Laser ablation ICP-MS was also used to visualize cisplatin distribution within a mouse cochlear tissue section, and the observed pattern of cisplatin distribution was similar to that seen in human tissue, with high signal intensity in the stria vascularis and low signal intensity in the organ of Corti across cochlear turns (Supplemental Fig. 3b). Since the basal turn of the cochlea is more susceptible to cisplatin-induced damage than the apical turn, we analyzed the relative cisplatin signal intensity within the stria vascularis in each turn (Rybak, Whitworth, Mukherjea, & Ramkumar, 2007). Cisplatin signal intensity was highest in the cochlear base, and it generally decreased with progression towards the apex (Supplemental Fig. 3c). Increased accumulation of cisplatin in the cochlear base is consistent with the increased susceptibility of that region to cisplatin-induced damage."

Methods, lines 299-300: "The entirety of each region from cochlear apex to base was isolated and analyzed as one."

4. Comment: Fig 1. Hearing loss (ABRs) appears greater in Recovery 1 (low frequency) than in recovery 2. This is not seen in DPOAE amplitudes.

Though the hearing loss as measured by auditory brainstem responses does trend worse at low frequencies (8 and 11.2 kHz) at the end of Recovery 1 than the end of Recovery 2, this difference is not statistically significant. However, at high frequencies, hearing loss *is* significantly greater at the end of Recovery 2 than Recovery 1. The DPOAE amplitudes are consistent with these findings, with similar amplitudes at low frequencies and decreased amplitudes at high frequencies for Recovery 2 compared to Recovery 1.

In response to the Reviewer's comment, the following has been added to the Results (lines 70-72):

"Hearing loss at the highest frequency (40 kHz) was significant after Recovery 2, while changes in lower frequencies reached significance after Cycle 3 (32, 22.4, and 8 kHz) or Recovery 3 (16 and 11.2 kHz)."

5. There appears to be temporal differences in loss of ABRs, DPOAE amplitudes and EP despite greater accumulation of cisplatin in the cochlea (especially stria vascularis) over the course of treatment. Any explanation?

As touched upon in the response above, the time course of changes in the ABRs and DPOAEs are consistent with one another. Both measures show minimal change at Recovery 1, high frequency deterioration at Recovery 2, and broad deterioration across frequencies at Recovery 3. Endocochlear potentials are the first physiologic measure to be significantly affected by cisplatin treatment, as they are significantly decreased at the end of Cycle 1. This is consistent with the idea that the stria vascularis is initially exposed to elevated levels of cisplatin and experiences both early and long-term damage.

In response to the Reviewer's comment, the manuscript has been updated to clarify the labeling of time points in Figure 3h, bringing them in line with the nomenclature used in Figures 1c & e.

6. The molecular weight of BODIPY FL cisplatin is about twice that of cisplatin. I wonder if this difference could partly explain its differential uptake in the organ of Corti versus stria vascularis via different transporters and/or ion channels.

Though BODIPY FL-cisplatin is approximately twice the molecular weight of cisplatin, previous work has demonstrated that cellular uptake of BODIPY-FL-cisplatin parallels that of cisplatin in other settings. For example, cisplatin-resistant cancer cells show reduced accumulation of BODIPY FL-cisplatin, just as they do for cisplatin (Jagodinsky et al., 2015). Our experiments using BODIPY FL-cisplatin are controlled for by comparison to the BODIPY FL-Boc ligand and are made in parallel with direct measurements of platinum by ICP-MS whenever possible. We see little accumulation of cisplatin-free BODIPY FL-Boc in the cochlea, suggesting that the BODIPY ligand doesn't drive uptake and retention of BODIPY FL-cisplatin. Our results obtained using BODIPY FL-cisplatin agree with those obtained by standard ICP-MS and laser ablation ICP-MS techniques in that all of these techniques demonstrate high accumulation of cisplatin in the stria vascularis and minimal accumulation in the organ of Corti. This further validates the use of BODIPY FL-cisplatin as a probe. The specific channels and/or transporters facilitating cellular uptake of BODIPY FL-cisplatin in the inner ear are not known, but those facilitating cisplatin uptake are also largely unknown, making it difficult to examine roles for particular channels/transporters at this time.

In response to the Reviewer's comment, the following has been added to the Results:

Lines 114-115: “[We sought to visualize the cellular localization of cisplatin in cochlea through use of the fluorescent cisplatin conjugate BODIPY FL-Cisplatin,] recently shown to have cellular uptake similar to cisplatin”

Lines 118-121: “[No specific BODIPY fluorescence signal was apparent in cochlear tissue from control mice treated with the unconjugated (cisplatin-free) dye BODIPY FL-Boc], suggesting that the BODIPY ligand was not responsible for the uptake and retention of BODIPY FL-cisplatin that we observed.”

Lines 123-126: “These data agree with those obtained by ICP-MS analysis of cochlear microdissections in suggesting that the stria vascularis accumulates the highest levels of cisplatin, and that cisplatin accumulation in mechanosensory hair cells is more limited.”

7. It is not clear whether the cisplatin detected by ICP-MS is active, could be readily activated or is sequestered by being covalently incorporated into antioxidant molecules, proteins or DNA. Thus, it is not clear whether this "retained" cisplatin could provide "active" cisplatin at levels high enough to produce subsequent (and progressive) hearing deficits.

As the ICP-MS technique quantifies platinum, it equally detects all forms of cisplatin, including bound and un-bound forms. Thus, we are unable to differentiate among different cisplatin species. However, this may be considered a strength of our approach, as the contributions of each of these forms of cisplatin to cellular toxicity within the cochlea are not known, and our approach allows for an unbiased quantification of total cisplatin species. The distinction between “active” (un-bound) and “retained” (bound) cisplatin is not straightforward, since cisplatin first retained in tissue may later be liberated. Slow dissociation of cisplatin from tissue collagen has recently been demonstrated (Chang et al., 2016), and long-term exposure to circulating platinum is associated with development of late side effects of cisplatin chemotherapy (Boer et al., 2015). Additionally, cisplatin bound to and sequestered by antioxidant molecules may nonetheless contribute to toxicity, since it depletes a cell’s effective pool of those molecules. Understanding the intracellular behavior of cisplatin represents an ongoing field of research important to both this field and the broader cancer field.

In response to the Reviewer’s comment, the following has been added to the Discussion (as well as overlapping changes in response to Comment 1 from Reviewer #2 which are noted below):

Lines 201-204: “Our ICP-MS based approach allowed for equal detection of bound and free (unbound) platinum, but not differentiation of the two. Further study of cisplatin reactivity with proteins and other biomolecules may further elucidate the exact mechanisms by which this drug causes cytotoxicity both within and outside the cochlea.”

Lines 196-198: “In support of this idea, cisplatin has recently been demonstrated to bind extensively to, and slowly dissociate from, type I collagen, the major protein component of bone (Chang et al., 2016).”

Reviewer #2 (Remarks to the Author):

1. However I cannot see the paper as being of interest to a wider field - it is specialized.

Though our study was performed with a focus upon cisplatin ototoxicity, our results hold broader implications for clinical cisplatin use. Outside the cochlea, we also find long-term retention of high levels of cisplatin in the long bones (femur). Long-term exposure to circulating platinum has previously been associated with a number of late toxicities of cisplatin treatment, and platinum has been detected in the blood of chemotherapy patients more than 20 years after treatment (Boer et al., 2015; Gietema et al., 2000; Hjelle et al., 2015; Sprauten et al., 2012; Tothill, Klys, Matheson, McKay, & Smyth, 1992). Such long-term elevated platinum levels in the blood are likely a result of cisplatin species being re-mobilized from a tissue reservoir. Our results suggest that bone, which represents a significant percentage of total body mass, may serve as a reservoir for a large quantity of cisplatin species, and may thus have an important role in mediating the late toxicities of this drug in cancer survivors.

In response to the Reviewer’s comment, the following has been added to the Discussion (lines 190-201):

“Outside the cochlea, we also found long-term retention of prominent levels of cisplatin in the long bones (femur). Long-term exposure to circulating platinum is associated with several late toxicities of cisplatin treatment, and platinum has been detected in the blood of chemotherapy patients more than 20 years after treatment (Boer et al., 2015; Gietema et al., 2000; Hjelle et al., 2015; Sprauten et al., 2012; Tothill et al., 1992). Such long-term elevated platinum levels in the blood are presumed to be a result of cisplatin species being re-mobilized from an unidentified tissue reservoir. Our results suggest that bone may serve as a reservoir for platinum, and prolonged release of platinum from bone may mediate the late toxicities of cisplatin in cancer survivors. In support of this idea, cisplatin has recently been demonstrated to bind extensively to, and slowly dissociate from, type I collagen, the major protein component of bone (Chang et al., 2016). Our results suggest that platinum may have pharmacokinetics akin to that of lead, another

heavy metal. In cases of toxicity, lead is known to distribute into bone, where it has a half-life of years-to-decades, and from which it can be exchanged back into the blood (Hu, Rabinowitz, & Smith, 1998). Lead toxicity is similarly associated with hearing loss (Osman et al., 1999).”

2. Another concern I have is that my understanding is that ototoxicity is a consequence of high-dose cisplatin. While I appreciate the technical difficulties and time involved it would be of interest to examine dose-dependence.

Though ototoxicity may be more severe with high dose cisplatin, it is certainly not limited to it. The limited number of studies that have been performed on patients receiving low-dose cisplatin regimens have demonstrated hearing loss in those patients as well, though more clinical study is needed (Schmitt & Page, 2017). Of note, existing clinical studies of low dose cisplatin ototoxicity have exclusively examined hearing loss at frequencies up to 4 or 8 kHz, and thus surely underestimate cisplatin-induced hearing loss, which affects even higher frequencies first. Though diverse cisplatin regimens are used to treat a wide variety of cancers, high prevalence of ototoxicity seems to be a shared adverse effect (Paken, Govender, Pillay, & Sewram, 2016).

In response to the Reviewer’s comment, the following has been added to the Discussion:

Lines 171-172: “Ototoxicity is a serious adverse effect of cisplatin chemotherapy, experienced by patients receiving either high or low-dose regimens (Cheraghi et al., 2015; Hitchcock, Tward, Szabo, Bentz, & Shrieve, 2009; Paken et al., 2016).”

Lines 182-184: “This audiometric monitoring should also include testing at frequencies up to 16 kHz, as extended range audiometry has been shown to improve the sensitivity for detecting cisplatin ototoxicity (Zuur et al., 2008).”

References:

- Boer, H., Proost, J. H., Nuver, J., Bunskoek, S., Gietema, J. Q., Geubels, B. M., . . . Gietema, J. A. (2015). Long-term exposure to circulating platinum is associated with late effects of treatment in testicular cancer survivors. *Ann Oncol*, 26(11), 2305-2310. doi:10.1093/annonc/mdv369
- Chang, Q., Ornatsky, O. I., Siddiqui, I., Straus, R., Baranov, V. I., & Hedley, D. W. (2016). Biodistribution of cisplatin revealed by imaging mass cytometry identifies extensive collagen binding in tumor and normal tissues. *Sci Rep*, 6, 36641. doi:10.1038/srep36641
- Cheraghi, S., Nikoofar, P., Fadavi, P., Bakhshandeh, M., Khoie, S., Gharehbagh, E. J., . . . Mahdavi, S. R. (2015). Short-term cohort study on sensorineural hearing changes in head and neck radiotherapy. *Med Oncol*, 32(7), 200. doi:10.1007/s12032-015-0646-3
- Gietema, J. A., Meinardi, M. T., Messerschmidt, J., Gelevert, T., Alt, F., Uges, D. R., & Sleijfer, D. T. (2000). Circulating plasma platinum more than 10 years after cisplatin treatment for testicular cancer. *Lancet*, 355(9209), 1075-1076.
- Hellberg, V., Wallin, I., Eriksson, S., Hernlund, E., Jerremalm, E., Berndtsson, M., . . . Laurell, G. (2009). Cisplatin and oxaliplatin toxicity: importance of cochlear kinetics as a determinant for ototoxicity. *J Natl Cancer Inst*, 101(1), 37-47. doi:10.1093/jnci/djn418
- Hitchcock, Y. J., Tward, J. D., Szabo, A., Bentz, B. G., & Shrieve, D. C. (2009). Relative contributions of radiation and cisplatin-based chemotherapy to sensorineural hearing loss in head-and-neck cancer patients. *Int J Radiat Oncol Biol Phys*, 73(3), 779-788. doi:10.1016/j.ijrobp.2008.05.040
- Hjelle, L. V., Gundersen, P. O., Oldenburg, J., Brydoy, M., Tandstad, T., Wilsgaard, T., . . . Haugnes, H. S. (2015). Long-term platinum retention after platinum-based chemotherapy in testicular cancer survivors: a 20-year follow-up study. *Anticancer Res*, 35(3), 1619-1625.

- Hu, H., Rabinowitz, M., & Smith, D. (1998). Bone lead as a biological marker in epidemiologic studies of chronic toxicity: conceptual paradigms. *Environ Health Perspect*, *106*(1), 1-8.
- Laurell, G., Andersson, A., Engstrom, B., & Ehrsson, H. (1995). Distribution of cisplatin in perilymph and cerebrospinal fluid after intravenous administration in the guinea pig. *Cancer Chemother Pharmacol*, *36*(1), 83-86. doi:10.1007/BF00685738
- Liu, H., Li, Y., Chen, L., Zhang, Q., Pan, N., Nichols, D. H., . . . He, D. Z. (2016). Organ of Corti and Stria Vascularis: Is there an Interdependence for Survival? *PLoS One*, *11*(12), e0168953. doi:10.1371/journal.pone.0168953
- Osman, K., Pawlas, K., Schutz, A., Gazdzik, M., Sokal, J. A., & Vahter, M. (1999). Lead exposure and hearing effects in children in Katowice, Poland. *Environ Res*, *80*(1), 1-8. doi:10.1006/enrs.1998.3886
- Paken, J., Govender, C. D., Pillay, M., & Sewram, V. (2016). Cisplatin-Associated Ototoxicity: A Review for the Health Professional. *J Toxicol*, *2016*, 1809394. doi:10.1155/2016/1809394
- Rybak, L. P., Whitworth, C. A., Mukherjea, D., & Ramkumar, V. (2007). Mechanisms of cisplatin-induced ototoxicity and prevention. *Hear Res*, *226*(1-2), 157-167. doi:10.1016/j.heares.2006.09.015
- Schmitt, N. C., & Page, B. R. (2017). Chemoradiation-induced hearing loss remains a major concern for head and neck cancer patients. *Int J Audiol*, 1-6. doi:10.1080/14992027.2017.1353710
- Sprauten, M., Darrah, T. H., Peterson, D. R., Campbell, M. E., Hannigan, R. E., Cvancarova, M., . . . Travis, L. B. (2012). Impact of long-term serum platinum concentrations on neuro- and ototoxicity in Cisplatin-treated survivors of testicular cancer. *J Clin Oncol*, *30*(3), 300-307. doi:10.1200/JCO.2011.37.4025
- Tothill, P., Klys, H. S., Matheson, L. M., McKay, K., & Smyth, J. F. (1992). The long-term retention of platinum in human tissues following the administration of cisplatin or carboplatin for cancer chemotherapy. *Eur J Cancer*, *28A*(8-9), 1358-1361.
- Zuur, C. L., Simis, Y. J., Verkaik, R. S., Schornagel, J. H., Balm, A. J., Dreschler, W. A., & Rasch, C. R. (2008). Hearing loss due to concurrent daily low-dose cisplatin chemoradiation for locally advanced head and neck cancer. *Radiother Oncol*, *89*(1), 38-43. doi:10.1016/j.radonc.2008.06.003

REVIEWERS' COMMENTS:

Reviewer #1 (Remarks to the Author):

The authors have adequately addressed my concerns. I recommend acceptance of this manuscript.

Reviewer #2 (Remarks to the Author):

The authors have revised their manuscript in line with reviewers comments. The analogies with lead toxicity are interesting and a new angle on the data. The work is convincing and well carried out - it is still essentially a bioanalytical paper. The findings will be of interest to the field and the question still remains how these findings could affect clinical treatment regimens. Secondly, the suggestions for therapeutic approaches based on these findings are still vague. I would recommend publication of this revision despite these caveats.

Reviewer #1 (Remarks to the Author):

The authors have adequately addressed my concerns. I recommend acceptance of this manuscript.

We thank the reviewer for thoughtful and constructive comments on the previous version of the manuscript.

Reviewer #2 (Remarks to the Author):

The authors have revised their manuscript in line with reviewers' comments. The analogies with lead toxicity are interesting and a new angle on the data. The work is convincing and well carried out - it is still essentially a bioanalytical paper. The findings will be of interest to the field and...

1. The question still remains how these findings could affect clinical treatment regimens.

We see potential for this work to affect clinical treatment regimens in two ways. First, as mentioned in the Discussion, our finding of long-term retention of cisplatin in the cochlea and others' previous findings of delayed hearing loss following cisplatin together suggest a need for long-term audiometric monitoring of cisplatin-treated patients. Consistent inclusion of audiometry as a component of long-term follow-up care after cisplatin chemotherapy would allow for the better identification of patients who could benefit from hearing aids or other hearing devices. Second, we believe our findings bring the field closer to developing an intervention which could be co-administered during cisplatin chemotherapy and act to prevent ototoxicity.

In response to the Reviewer's comment, the following has been added to the Discussion (lines 181-184):

“These findings suggest that hearing testing should be consistently included as a component of long-term follow-up care after cisplatin chemotherapy in order to identify all patients who could benefit from hearing rehabilitation.”

Additional relevant changes to the Discussion are detailed directly below.

2. Secondly, the suggestions for therapeutic approaches based on these findings are still vague.

Our findings suggest that inhibition of cisplatin uptake into the cochlea may be a more promising approach than trying to counteract the toxicity of cisplatin already in the cochlea. We have updated the Discussion in order to better clarify the type of therapy, and development thereof, suggested by our study.

In response to the Reviewer's comments 1 & 2, the following has been added to the Discussion (lines 219-224):

“[Our results point to a strategy aimed at prevention of cisplatin uptake into the stria vascularis as a promising therapeutic approach to prevention of cisplatin ototoxicity.] The mouse model and ICP-MS techniques detailed here will allow for the screening of bioactive molecules that may reduce the uptake of cisplatin into the cochlea. Local administration of such a drug could prevent ototoxicity without interfering with the anti-cancer efficacy of cisplatin.”

I would recommend publication of this revision despite these caveats.

We thank the reviewer for thoughtful and constructive comments on the manuscript.